# Effects of *jogi, Micropogonias undulatus,* addition on the production of volatile compounds in *baechu-kimchi*

**Gawon Lee**[1], **Sojeong Heo**[1], **Junghyun Park**[1], **Jung-Sug Lee**[2], **Do-Won Jeong** [1]*

**1** Department of Food and Nutrition, Dongduk Women's University, Seoul, Republic of Korea, **2** Department of Food and Nutrition, Kookmin University, Seoul, Republic of Korea

* jeongdw@dongduk.ac.kr

**Data Availability Statement:** All relevant data are within the manuscripts.

**Funding:** This work was supported by a grant (RS-2022-IP322014) of Korea Institute of Planning and Evaluation for Technology in Food, Agriculture and

## Abstract

*Baechu-kimchi* is a traditional vegetable fermented food using kimchi cabbage (*Brassica rapa*) as the main ingredient. A wide variety of ingredients can be used in kimchi depending on the specific region and even household. Although there have been a lot of studies examining various aspects of kimchi, there has been limited research on kimchi with added fish as a minor ingredient. Therefore, in the present work we aimed to assess changes in the volatile compounds of *baechu-kimchi* with the addition of seafood used as minor ingredients of kimchi. Sulfur compounds were the most commonly detected volatile compounds; 9 categories of volatile components were detected in total. Altogether, 30 sulfur compounds were detected, and among them, five sulfur compounds: (*E*)-1-(methyltrisulfanyl)prop-1-ene, 1-(methyldisulfanyl)-1-methylsulfanylpropane, (methyltetrasulfanyl)methane, 1-(methyldisulfanyl)-1-[(*E*)-prop-1-enyl]sulfanylpropane, and 1,1-bis(methyldisulfanyl)propane, were found only in *jogi*-added kimchi, thus confirming the influence of *jogi* addition. Principal component analysis revealed clear distinctions in the volatile compounds as a result of *jogi* addition as fermentation progressed. Moreover, when confirming the correlation with microbial populations, it was evident that the differentiation in volatile compounds was more attributable to *jogi* addition than microbial impact. In conclusion, the addition of *jogi* to *baechu-kimchi* led to an abundance of volatile compounds by the 20th day of fermentation.

## Introduction

Kimchi using *baechu* as a major ingredient is a naturally fermented food for which Codex International Food Standards (CODEX STAN 223–2001) were established in July 2001; *baechu* is kimchi cabbage (CODEX classification No. VB 0467) of the *Brassica rapa* vegetables [1]. *Baechu-kimchi* made with kimchi cabbage varies according to both the minor ingredients used and the region [2, 3]. For example, in coastal areas, seafood is typically added as a minor ingredient when making kimchi [4–7].

Kimchi is a fermented food that features rich tastes and different textures that are imparted through fermentation from raw materials such as *baechu* [8]. During the fermentation process,

Forestry (IPET) through the High Value-added Food Technology Development Program funded by the Ministry of Agriculture, Food and Rural Affairs (MAFRA), Republic of Korea. The funders had no role in study design, data collection and analysis, decision to publish, or preparation of the manuscript.

**Competing interests:** The authors have declared that no competing interests exist.

kimchi forms bacterial communities, which change the ingredients of the raw materials because of the enzymes and metabolites they produce [9–12]. The main bacteria formed in fermented kimchi are lactic acid bacteria, which produce organic acids, amino acids, or volatile compounds during the fermentation process. Lactic acid bacteria also contribute to the sensory properties of fermented *baechu-kimchi* [13, 14]. During the appropriate ripening period when kimchi is properly fermented, it has excellent sensory properties, but after the appropriate ripening period, the sour taste becomes stronger, the tissue becomes softer, and the composition or content of the volatile ingredients changes [8, 15, 16].

To date, there have been several studies reporting on the volatile compounds of kimchi. These include studies examining changes in the volatile compounds of long-term fermented *baechu-kimchi* [17], changes in the volatile compounds of mustard leaf (*Brassica juncea*) kimchi [18], changes in the volatile compounds of *Angelica keiskei* kimchi [19], changes in the volatile compounds of *Gamdongchotmoo* kimchi [20], changes in the volatile odor components of kimchi as a result of heating [21], and changes in the volatile flavor compounds in kimchi due to the addition of different ingredients (green onions, garlic, ginger, red pepper powder) [22]. Although there has been a study reporting on the effect that the addition of salted fish has on volatile compounds, there has been no research examining the effect on volatile compounds due to the addition of seafood [23].

In previous studies, we confirmed the effect that the addition of *jogi*—which is typically added into kimchi in Gyeongsangbuk-do—had on the bacterial communities of kimchi [24, 25]. Contrary to our assumptions that the protein content of fish would lead to many protein-using bacteria, there was no significant change in bacterial communities during kimchi fermentation. Other experiments also confirmed that the addition of seafood, *galchi* (*Trichiurus lepturus*), *gul* (*Crassostrea gigas*), *jeonbok* (*Nordotis discus discus*), *hongeu* (*Okamejei kenojei*), and *myeongtae* (*Theragra chalcogramma*), did not affect the microbial communities of kimchi [24, 26]. Another study showed that the addition of *jogi* into kimchi did not have significant effects on organic acid production [27]; however, the amino acid content in the early stages of fermentation was different. It was confirmed that the contents of glutamic acid and aspartic acid, which are savory ingredients, were significantly different immediately after the kimchi was prepared [27]. Altogether, the results confirmed that *jogi* addition did not significantly affect organic acids or bacterial communities, but that it did affect amino acid content. Nevertheless, there has still been insufficient research examining the effect of *jogi* addition on the sensory properties of kimchi. Therefore, this experiment was intended to confirm the effect of *jogi* addition on the volatile compounds that are produced during kimchi fermentation.

## Materials and methods

### Kimchi samples

The same kimchi that was used in the previous experiment [24, 25, 27] was used in the current work. Briefly, it is a naturally fermented *baechu-kimchi* (control group) that is manufactured in Sangju, Gyeongsangbuk-do, Republic of Korea; *jogi-baechu-kimchi* was this kimchi with the addition of *jogi* to approximately 5% of the total volume. Kimchi was fermented over 20 days at 10°C after the initial preparation. Kimchi was sampled every 5 days and stored -80°C until it was analyzed. In a previous study, there was a significant change in the microbial community on 10th day of fermentation [24, 25], and a notable difference in amino acids was observed on 20th day of fermentation [27]. Therefore, for the analysis of volatile compounds, kimchi samples taken on day 0 (immediately after preparation), day 10, and day 20 were used.

### Analysis of volatile compounds of kimchi

One gram of kimchi sample was mixed with 3.25 mL of water and 0.15 g of NaCl in a 15 mL vial with a tan polytetrafluoroethylene/silicone septum (tan PTFE/silicone septum; Supelco, Bellefonte, PA, USA). The sample prepared in this way was then kept at 60˚C for 30 min. The volatile compounds were absorbed onto a 50/30 μm divinylbenzene/carboxen/polydimethylsiloxane fiber (DVB/CAR/PDMS fiber; Supelco) for 15 min, and then eluted at 220˚C. The volatile compounds were transferred through the MS transfer line at 230˚C (ISQ 7000™ Single Quadrupole GC-MS system; Thermo Fisher Scientific, USA). Separation was performed on a TG-WAXMS GC column (30 m length × 0.25 mm i.d. × 0.25 μm film thickness; Thermo Fisher Scientific). The oven program was held at 40˚C for 3 min, raised at 2˚C/min to 150˚C and then held at 150˚C for 10 min, and raised again at 4˚C/min to 200˚C and held at 200˚C for 10 min; the carrier gas (He) flow rate was 1.0 mL/min, ionization energy was 70 eV, and the mass scan range was 50–550 $m/z$. Chromeleon 7.1 Chromatography Data System Software was used to pre-process raw GC-MS data (Dionex, Sunnyvale, CA, USA).

The internal standard was methyl cinnamate (99%) solution at 1.5 μL of 1,000 ppm. The quantitative data were acquired by comparing the peak area (min x counts) of the experimental value obtained by including 1-fold, 2-fold, and 3-fold internal standards excluding kimchi samples, represented by a linear equation. The analysis of volatile compounds was conducted in triplicate for each sample.

### Correlation between volatile compounds, microbial community, and fermentation period

PAST 4.10 software was used to visualize the correlation between the fermentation period and volatile compounds, and the microbial community based on culture dependent analysis [24, 25] and volatile compounds of kimchi. Spearman's rank correlation was used, and values with $p < 0.05$ were considered to be statistically significant.

### Statistical analysis

Duncan's multiple range test following a one-way analysis of variance (ANOVA) was used to evaluate significant differences between average values of volatile compounds. Values with $p < 0.05$ were considered to be statistically significant. To visualize differences between volatile compounds of both kimchi samples, principal component analysis (PCA) was performed using SPSS software v.27 (SPSS Inc., Chicago, IL, USA). Additionally, Pearson correlation coefficient ($r$) was also calculated using SPSS software.

## Results and discussion

### Composition of volatile compounds in kimchi fermentation

To investigate the impact of *jogi* addition on the production of volatile compounds during the fermentation process of kimchi, volatile compounds were analyzed by GC-MS through SPME extraction on different fermentation days (Fig 1). Kimchi without the addition of *jogi*, *baechu-kimchi*, was used as the control group. The evaluations of total acidity, pH, and microbial count, which are indicators of successful fermentation, were established in a prior experiment [24, 25]. On the first day of fermentation for the control group, seven types of volatile compounds were identified in total, and as fermentation progressed, acid and isoprenoid compounds were newly added (Fig 1A). The kimchi with *jogi* addition, *jogi-baechu-kimchi*, showed a similar trend, but ketone volatile compounds were maintained on the 20th day of

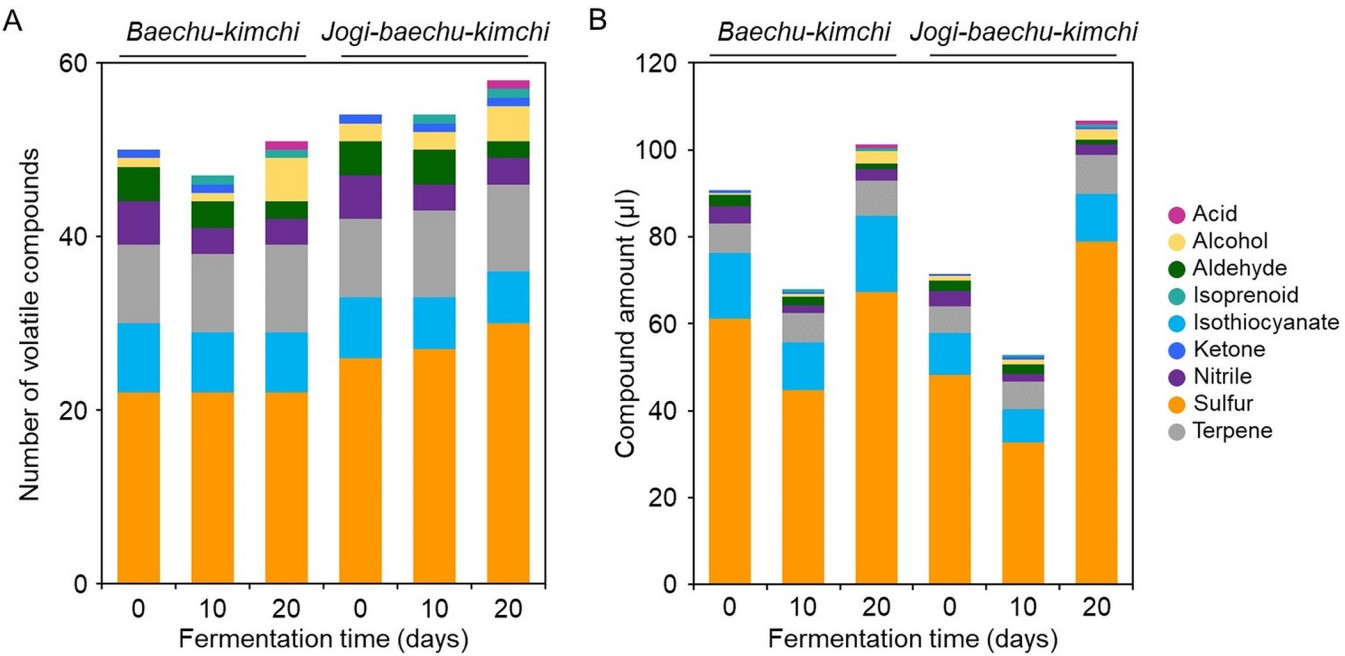

**Fig 1.** Number of volatile compounds (A), and amounts of compounds (B) identified in *baechu-kimchi* and *jogi-baechu-kimchi*. The following number indicates the fermentation time (in days).

fermentation. It was also observed that there were more volatile compounds related to sulfur in the *jogi*-added kimchi.

Although the number of volatile compounds is higher in *jogi-baechu-kimchi*, when they were quantified, the amounts of volatile compounds were higher in *baechu-kimchi* until the 10th day of fermentation (Fig 1B). On the first day of fermentation, alcohols, aldehydes, isothiocyanates, ketones, nitriles, sulfur compounds, and terpenes were detected, with the amounts of all compounds aside from alcohol being the same or higher in the *baechu-kimchi* than they were in both kimchi types. On the 10th day of fermentation, isoprenoid compounds not detected on day 0 were also identified. Comparing the 10th day to the day 0, six categories, excluding alcohol and aldehyde, showed similar or higher compound amounts in the *baechu-kimchi*. However, on the 20th day of fermentation, *jogi-baechu-kimchi* displayed higher amounts of volatile compounds, with a particularly high level of sulfur compound detected. Sulfur compounds are known to be major volatile compounds in fish fermentation, and they are classified as strong-smelling substances because of their low odor threshold; moreover, their quantity increases during fish fermentation [28–30]. On the 20th day of fermentation, additional acid compounds were detected compared with the findings on the 10th day, and in the *baechu-kimchi*, ketone compounds disappeared. Furthermore, on the 20th day of fermentation, only sulfur compounds and terpene compounds were found in higher amounts in *jogi-baechu-kimchi*.

## Effect of *jogi* addition on the production of volatile compounds

The volatile compounds detected in *baechu-kimchi* and *jogi-baechu-kimchi* were individually examined (Table 1). To investigate changes that occurred based on the presence or absence of *jogi* addition, the analysis focused on the differences between the two types of kimchi.

Acetic acid was detected in the acid category, and this was observed in both types of kimchi on the 20th day of fermentation. Acetic acid, which is an organic acid that is known to impart

**Table 1. Effects of *jogi* addition of *baechu-kimchi* on the production of volatile compounds.**

| Volatile compounds | Abbreviation | RT | RI | *Baechu-kimchi* | | | *Jogi-baechu-kimchi* | | |
|---|---|---|---|---|---|---|---|---|---|
| | | | | 0 Day | 10 Day | 20 Day | 0 Day | 10 Day | 20 Day |
| Acid | | | | | | | | | |
| Acetic acid | AC1 | 26.96 | 1449 | | | 0.935[c,A] | | | 0.773[b,A] |
| Alcohol | | | | | | | | | |
| Hexan-1-ol | ALC1 | 20.72 | 1355 | | | 0.606[c,A] | 0.537[b,B] | 0.522[b,B] | 0.596[c,A] |
| 6-Methylhept-5-en-2-ol | ALC2 | 27.38 | 1465 | | | 0.569[c,A] | | | 0.552[b,A] |
| (*E*)-Oct-2-en-1-ol | ALC3 | 35.89 | 1614 | | | 0.560[b,A] | | | 0.578[c,A] |
| 2-Phenylethanol | ALC4 | 51.25 | 1906 | | | 0.569[b,A] | | | |
| 4-Methyl-3-propan-2-yldec-1-en-4-ol | ALC5 | 75.69 | 1366 | 0.534[ab,A] | 0.531[ab,A] | 0.545[b,A] | 0.527[a,A] | 0.527[a,A] | 0.532[ab,A] |
| Aldehyde | | | | | | | | | |
| (*E*)-Hex-2-enal | ALD1 | 12.87 | 1216 | 0.913[c,A] | 0.848[c,A] | | 0.777[bc,A] | 0.560[b,A] | |
| (*E*)-Hept-2-enal | ALD2 | 18.67 | 1323 | 0.536[b,A] | | 0.618[d,A] | 0.529[b,A] | 0.523[b,B] | 0.567[c,B] |
| Nonanal | ALD3 | 22.87 | 1391 | 0.555[ab,A] | 0.551[ab,A] | 0.584[ab,A] | 0.540[a,A] | 0.532[a,A] | 0.608[b,A] |
| (2*E*,4*E*)-Hepta-2,4-dienal | ALD4 | 28.53 | 1495 | 0.586[c,A] | 0.589[c,A] | | 0.557[bc,A] | 0.540[b,A] | |
| Isoprenoid | | | | | | | | | |
| (*E*)-4-(2,6,6-Trimethylcyclohexen-1-yl)but-3-en-2-one | I1 | 52.19 | 1940 | | 0.612[b,A] | 0.695[c,A] | | 0.584[b,A] | 0.635[b,A] |
| Isothiocyanate | | | | | | | | | |
| 3-Isothiocyanatoprop-1-ene | IS1 | 21.08 | 1356 | 1.902[c,A] | 1.920[c,A] | 0.909[ab,A] | 1.583[bc,A] | 0.736[a,A] | 0.918[ab,A] |
| 4-Isothiocyanatobut-1-ene | IS2 | 26.55 | 1435 | 5.334[a,A] | 5.151[a,A] | 10.403[b,A] | 3.826[a,A] | 3.742[a,A] | 6.063[a,A] |
| 1-Isothiocyanatononane | IS3 | 34.06 | 1862 | | 0.572[b,A] | 0.618[c,A] | | | |
| 1-Isothiocyanatoheptane | IS4 | 39.97 | 1666 | 0.543[c,A] | | 0.571[d,A] | | 0.523[b,B] | |
| 1-Isothiocyanato-3-methylsulfanylpropane | IS5 | 54.33 | 1975 | 0.568[d,A] | 0.521[b,A] | | 0.546[c,A] | | 0.529[bc,B] |
| 1-Isothiocyanato-4-methylsulfanylbutane | IS6 | 61.60 | 2087 | 1.578[b,A] | 0.536[a,A] | 0.588[a,A] | 0.906[a,A] | 0.566[a,A] | 0.604[a,A] |
| (*Z*)-4-Isothiocyanato-1-methylsulfanylbut-1-ene | IS7 | 61.94 | 1419 | 0.707[c,A] | | | 0.576[b,B] | | |
| 2-Isothiocyanatoethylbenzene | IS8 | 67.42 | 2234 | 3.464[b,A] | 1.626[a,A] | 3.675[b,A] | 1.524[a,B] | 1.472[a,A] | 2.070[a,A] |
| 1-Isothiocyanato-5-methylsulfanylpentane | IS9 | 69.96 | 2242 | 1.088[b,A] | 0.561[a,A] | 0.657[a,A] | 0.774[a,A] | 0.710[a,A] | 0.668[a,A] |
| Ketone | | | | | | | | | |
| Undecan-2-one | K1 | 34.75 | 1598 | 0.543[bc,A] | 0.535[b,A] | | 0.535[b,A] | 0.527[b,A] | 0.562[c,B] |
| Nitrile | | | | | | | | | |
| Hex-5-enenitrile | N1 | 20.03 | 858 | 0.661[a,A] | 0.602[a,A] | 0.930[b,A] | 0.672[a,A] | 0.562[a,A] | 0.830[b,A] |
| 5-Methylsulfanylpentanenitrile | N2 | 51.87 | 1851 | 0.541[b,A] | | | 0.536[b,A] | | |
| 3-(Thiiran-2-yl)propanenitrile | N3 | 52.20 | 1006 | 0.754[b,A] | | | 0.720[b,A] | | |
| 3-Phenylpropanenitrile | N4 | 57.11 | 2041 | 1.428[c,A] | 0.714[a,A] | 1.160[bc,A] | 0.976[ab,A] | 0.723[a,A] | 0.983[ab,A] |
| 6-Methylsulfanylhexanenitrile | N5 | 58.01 | 2089 | 0.614[c,A] | 0.534[a,A] | 0.547[ab,A] | 0.591[bc,A] | 0.556[ab,A] | 0.588[bc,A] |
| Sulfur | | | | | | | | | |
| 3-Methylsulfanylprop-1-ene | S1 | 5.21 | 956 | 0.595[bc,A] | 0.583[bc,A] | | 0.573[bc,A] | 0.549[b,A] | 0.650[c,B] |
| (Methyldisulfanyl)methane | S2 | 6.71 | 1077 | 5.390[c,A] | 1.681[a,A] | 0.997[a,A] | 3.747[b,A] | 1.333[a,A] | 1.473[a,B] |
| 3-Prop-2-enylsulfanylprop-1-ene | S3 | 10.06 | 1148 | 1.095[ab,A] | 1.354[b,A] | 1.908[c,A] | 0.889[ab,A] | 0.790[a,A] | 2.222[c,A] |
| 1-(Methyldisulfanyl)propane | S4 | 13.38 | 1239 | 0.686[a,A] | 0.786[ab,A] | 0.948[b,A] | 0.664[a,A] | 0.623[a,A] | 1.275[c,B] |
| 3,4-Dimethylthiophene | S5 | 14.75 | 1252 | | 0.562[bc,A] | | 0.554[bc,B] | 0.527[b,B] | 0.586[c,B] |
| (*E*)-1-(Methyldisulfanyl)prop-1-ene | S6 | 15.06 | 1327 | 0.854[d,A] | 0.660[abc,A] | 0.618[ab,A] | 0.786[cd,A] | 0.574[a,A] | 0.765[bcd,A] |
| 3-(Methyldisulfanyl)prop-1-ene | S7 | 16.12 | 1281 | 7.940[c,A] | 4.532[ab,A] | 5.623[bc,A] | 5.962[bc,A] | 2.142[a,A] | 7.206[bc,A] |
| (Methyltrisulfanyl)methane | S8 | 21.41 | 1377 | 4.849[b,A] | 2.781[ab,A] | 4.339[b,A] | 4.293[b,A] | 2.119[a,A] | 4.697[b,A] |
| 2-(Prop-2-enyldisulfanyl)propane | S9 | 24.64 | 1076 | 0.816[a,A] | 1.214[ab,A] | 1.700[b,A] | 0.771[a,A] | 0.723[a,A] | 2.364[c,A] |
| 1-[[(*E*)-Prop-1-enyl]disulfanyl]propane | S10 | 25.02 | 1118 | 0.575[a,A] | 0.570[a,A] | 0.575[a,A] | 0.557[a,A] | 0.522[a,A] | 0.699[b,B] |
| 3-[(Prop-2-en-1-yl)disulfanyl]prop-1-ene | S11 | 27.78 | 1475 | 9.040[a,A] | 10.723[ab,A] | 17.538[bc,A] | 7.374[a,A] | 5.103[a,A] | 19.582[c,A] |
| (*E*)-1-(Prop-2-enyldisulfanyl)prop-1-ene | S12 | 28.04 | 1103 | 4.656[bc,A] | 2.487[ab,A] | 3.381[abc,A] | 2.944[abc,A] | 1.463[a,A] | 5.110[c,A] |

(*Continued*)

**Table 1.** (Continued)

| Volatile compounds | Abbreviation | RT | RI | *Baechu-kimchi* | | | *Jogi-baechu-kimchi* | | |
|---|---|---|---|---|---|---|---|---|---|
| | | | | 0 Day | 10 Day | 20 Day | 0 Day | 10 Day | 20 Day |
| 3H-1,2-Dithiole | S13 | 29.89 | 1510 | 0.978[c,A] | 0.756[a,A] | 0.795[ab,A] | 0.779[ab,A] | 0.621[a,B] | 0.952[bc,A] |
| 1-(Methyltrisulfanyl)propane | S14 | 30.01 | 1531 | | | 0.875[c,A] | 0.600[b,B] | 0.566[b,B] | 0.989[c,A] |
| 4-Ethyl-5-methyl-1,3-thiazole | S15 | 31.15 | 1412 | 11.168[abc,A] | 7.757[ab,A] | 17.390[c,A] | 5.278[a,A] | 6.269[a,A] | 13.655[bc,A] |
| 3-(Methyltrisulfanyl)prop-1-ene | S16 | 33.66 | 1593 | 6.273[bc,A] | 3.699[ab,A] | 4.902[abc,A] | 5.760[bc,A] | 2.395[a,A] | 7.090[c,A] |
| (Z)-1-(Methyltrisulfanyl)prop-1-ene | S17 | 34.14 | 1179 | 0.617[abc,A] | 0.570[ab,A] | 0.593[abc,A] | 0.652[bc,A] | 0.543[a,A] | 0.680[c,A] |
| (E)-1-(Methyltrisulfanyl)prop-1-ene | S18 | 34.33 | 1586 | | | | | 0.539[b,B] | 0.654[c,B] |
| (Methyldisulfanyl)-methylsulfanylmethane | S19 | 37.53 | 1650 | 0.667[c,A] | 0.532[a,A] | 0.538[a,A] | 0.602[b,A] | 0.542[a,A] | 0.545[a,A] |
| 1,2,5,6,7,8-Hexahydropyrrolizine-3-thione | S20 | 37.84 | 1222 | 0.612[a,A] | 0.575[a,A] | 0.625[a,A] | 0.572[a,A] | 0.573[a,A] | 0.607[a,A] |
| 1-(Propyltrisulfanyl)propane | S21 | 37.94 | 1703 | | | 0.560[b,A] | | | 0.563[b,A] |
| 1-(Methyldisulfanyl)-1-methylsulfanylpropane | S22 | 40.84 | 1207 | | | | | | 0.559[b,B] |
| 3-Ethenyl-3,6-dihydrodithiine | S23 | 41.34 | 1750 | 0.715[b,A] | 0.561[a,A] | 0.608[ab,A] | 0.621[ab,A] | 0.533[a,A] | 0.611[ab,A] |
| (Methyltetrasulfanyl)methane | S24 | 41.63 | 1746 | | | | 0.584[d,B] | 0.530[b,B] | 0.550[c,B] |
| 3-(Prop-2-enyltrisulfanyl)prop-1-ene | S25 | 44.52 | 1805 | 1.674[bc,A] | 1.182[ab,A] | 1.514[abc,A] | 1.412[abc,A] | 0.838[a,A] | 1.959[c,A] |
| 2-Ethenyl-4H-1,3-dithiine | S26 | 47.15 | 1857 | 0.737[b,A] | 0.659[ab,A] | 0.768[b,A] | 0.646[ab,A] | 0.583[a,A] | 0.747[b,A] |
| Methylsulfanyl(methylsulfinyl)methane | S27 | 48.41 | 1042 | 0.566[d,A] | | | 0.541[c,A] | 0.522[b,B] | 0.535[bc,B] |
| 1-(Methyldisulfanyl)-1-[(E)-prop-1-enyl]sulfanylpropane | S28 | 48.83 | 1413 | | | | | 0.523[b,B] | 0.550[c,B] |
| 1,1-Bis(methyldisulfanyl)propane | S29 | 56.65 | 1483.4 | | | | 0.531[b,B] | | 0.536[b,B] |
| 1,3-Dithiole-2-thione | S30 | 79.14 | 1172 | 0.547[b,A] | 0.526[a,A] | 0.550[b,A] | 0.524[a,B] | 0.521[a,A] | 0.551[b,A] |
| Terpene | | | | | | | | | |
| 3,7-Dimethylocta-1,6-dien-3-ol | T1 | 32.35 | 1547 | 0.568[a,A] | 0.570[a,A] | 0.686[b,A] | 0.551[a,A] | 0.546[a,A] | 0.657[b,A] |
| (1,7,7-Trimethyl-2-bicyclo[2.2.1]heptanyl) acetate | T2 | 33.20 | 1277 | 0.538[b,A] | 0.530[ab,A] | 0.535[ab,A] | 0.535[ab,A] | 0.522[a,A] | 0.543[b,A] |
| 2,6,6-Trimethylcyclohexene-1-carbaldehyde | T3 | 35.28 | 1611 | | | 0.734[b,A] | | | |
| 1-Methyl-4-(6-methylheptan-2-yl)benzene | T4 | 38.39 | 1696 | | | | | 0.527[b,B] | 0.555[c,B] |
| (2Z)-3,7-Dimethylocta-2,6-dienal | T5 | 39.10 | 1680 | 0.606[c,A] | 0.547[ab,A] | 0.567[ab,A] | 0.575[bc,A] | 0.534[a,A] | 0.582[bc,A] |
| 2-Methyl-5-(6-methylhept-5-en-2-yl)bicyclo[3.1.0]hex-2-ene | T6 | 41.24 | 1384 | 0.908[ab,A] | 1.196[b,A] | 0.821[ab,A] | 0.752[a,B] | 0.929[ab,A] | 1.651[c,B] |
| (4S)-1-Methyl-4-(6-methylhepta-1,5-dien-2-yl)cyclohexene | T7 | 41.54 | 1727 | 0.941[bcd,A] | 0.774[abc,A] | 1.064[d,A] | 0.742[ab,B] | 0.654[a,A] | 0.956[cd,A] |
| (3E,6E)-3,7,11-Trimethyldodeca-1,3,6,10-tetraene | T8 | 43.17 | 1746 | 0.644[ab,A] | 0.639[ab,A] | 0.723[bc,A] | 0.595[a,A] | 0.593[a,A] | 0.739[c,A] |
| 3-(6-Methylhept-5-en-2-yl)-6-methylidenecyclohexene | T9 | 43.74 | 1772 | 0.988[ab,A] | 0.988[ab,A] | 1.233[bc,A] | 0.901[ab,A] | 0.759[a,A] | 1.354[c,A] |
| 1-Methyl-4-(6-methylhept-5-en-2-yl)benzene | T10 | 44.08 | 1777 | 0.920[ab,A] | 0.943[ab,A] | 1.018[bc,A] | 0.796[ab,B] | 0.740[a,A] | 1.209[c,A] |
| (2E)-3,7-Dimethylocta-2,6-dien-1-ol | T11 | 48.52 | 1847 | 0.705[ab,A] | 0.618[a,A] | 0.801[b,A] | 0.647[a,A] | 0.585[a,A] | 0.788[b,A] |

[1] a, b, c, d Different superscripts in a column indicate significant differences at $p < 0.05$ according to independent *t*-test.

[2] A, B Different superscripts in a row indicate significant differences at $p < 0.05$ according to Duncan's multiple range test.

a sour taste to kimchi, has also been observed to increase in quantity as fermentation progressed in previous experiments [31]. Therefore, it can be inferred that this volatile compound is produced during the fermentation process of kimchi and is not influenced by *jogi* addition.

In the alcohol category, five volatile compounds were detected: hexan-1-ol, 6-methylhept-5-en-2-ol, (*E*)-oct-2-en-1-ol, 2-phenylethanol, and 4-methyl-3-propan-2-yldec-1-en-4-ol. Aside from 2-phenylethanol, there were no significant differences in the detected amounts of the compounds, indicating that they are commonly produced during kimchi fermentation. Interestingly, 2-phenylethanol was detected in the control group but not in the *jogi-baechu-kimchi*. Previous studies have shown findings of 2-phenylethanol detection from day 0 to day 30 of fermentation regardless of the addition of fish sauce [32]. However, recent analyses of kimchi using the same SPME fiber (DVB/CAR/PDMS, 50/30 μm, Supelco) did not confirm the presence of 2-phenylethanol [33]. It is therefore speculated that the absence of this volatile

compound in *jogi-baechu-kimchi* may be due to sample variations rather than a reduction caused by *jogi* addition, or that it may be a component that is detected later in the fermentation process.

Four volatile compounds were detected in the aldehyde category, (*E*)-hex-2-enal, (*E*)-hept-2-enal, nonanal, and (2*E*,4*E*)-hepta-2,4-dienal, while in the isoprenoid category, one volatile compound, (*E*)-4-(2,6,6-trimethylcyclohexen-1-yl)but-3-en-2-one, was identified. However, the only statistically significant difference was observed in (*E*)-hept-2-enal, which was found to be higher in *baechu-kimchi*. (*E*)-Hex-2-enal is frequently detected in kimchi fermentation [32], and the decrease in the quantity of (*E*)-hex-2-enal is presumed to be due to the addition of *jogi* affecting this volatile compound.

In the isothiocyanate category, 1-isothiocyanatononane was detected from the mid-fermentation stage in the control group, while (*Z*)-4-isothiocyanato-1-methylsulfanylbut-1-ene was observed only on the first day of fermentation and then disappeared, irrespective of the addition of *jogi*. Isothiocyanates are pungent-odor volatile compounds that are typically detected in the early stages of fermentation, but which tend to decrease as fermentation progresses [34]. A similar trend was observed in this experiment, with the number of isothiocyanates in both *baechu-kimchi* and *jogi-baechu-kimchi* decreasing as fermentation progressed compared with that in the early fermentation stages.

In the ketone category, undecan-2-one was detected. It was present in *baechu-kimchi* until the 10th day, while in *jogi-baechu-kimchi*, the quantity remained similar even on the 20th day. A more thorough investigation as to whether the addition of *jogi* prolongs the presence of the undecan-2-one is needed.

The sulfur compound exhibited the highest number of volatile compounds. These compounds are known to be generated from the main and minor ingredients used when making kimchi, such as cabbage, radish, red pepper, garlic, green onion, and ginger [35]. In total, 20 sulfur compounds were consistently detected in all samples regardless of the presence of *jogi* addition. However, five sulfur compounds, namely (*E*)-1-(methyltrisulfanyl)prop-1-ene, 1-(methyldisulfanyl)-1-methylsulfanylpropane, (methyltetrasulfanyl)methane, 1-(methyldisulfanyl)-1-[(*E*)-prop-1-enyl]sulfanylpropane, and 1,1-bis(methyldisulfanyl)propane, were exclusively found in *jogi-baechu-kimchi*. Among these, the four volatile compounds that remain after excluding (methyltetrasulfanyl)methane are also detected in garlic and onion [36–39]. Regarding (methyltetrasulfanyl)methane, it has been identified as a representative volatile compound in shrimp paste and fish [40, 41], suggesting that it might be derived from seafood.

In the terpene category, nine volatile compounds were detected in all samples, regardless of the fermentation period or *jogi* addition. 2,6,6-Trimethylcyclohexene-1-carbaldehyde was found only on the 20th-day for *baechu-kimchi*, while 1-methyl-4-(6-methylheptan-2-yl)benzene was detected in the 10 and 20th-day of *jogi-baechu-kimchi*. Terpenes have previously been detected in large quantities in the early stages of kimchi fermentation, where they have also been shown to decrease after 30 days of fermentation [32]. However, in this experiment, kimchi fermentation was only conducted for 20 days, and there was no significant difference in terpene content. Moreover, sesquiterpenes are relatively less influential in the flavor of kimchi compared with sulfur compounds because of their higher odor threshold levels [32].

The above results confirm that *jogi*-addition affects the production of sulfur compounds among volatile compounds. Specifically, five volatile compounds, (*E*)-1-(methyltrisulfanyl)prop-1-ene, 1-(methyldisulfanyl)-1-methylsulfanylpropane, (methyltetrasulfanyl)methane, 1-(methyldisulfanyl)-1-[(*E*)-prop-1-enyl]sulfanylpropane, and 1,1-bis(methyldisulfanyl)propane, are observed exclusively in *jogi-baechu-kimchi*, indicating the influence of *jogi*. The volatile compounds detected only in the *jogi-baechu-kimchi* may have been released from the *jogi* during the fermentation period, or they could have been produced from proteins by

microorganisms or enzymes originally present in the *jogi*. Further in-depth research on these compounds is suggested. Nonetheless, the key finding from this study is that the addition of *jogi* influenced the volatile compound profile of the kimchi.

## PCA and screening of volatile profiles of kimchi fermentation

To visually assess the types and quantities of volatile compounds produced in *jogi-baechu-kimchi* and *baechu-kimchi*, principal component analysis (PCA) was conducted using SPSS (Fig 2). Among the nine volatile compounds categories, the most detected sulfur compounds are generally distributed, primarily in the positive position based on PC1 (Fig 2A). The volatile compounds produced in both types of kimchi were found to be positioned in the first or second quadrant on day 0 and then shifted to the second or third quadrant on the 10th day, thus indicating a negative correlation with PC1 (Fig 2B). However, *jogi-baechu-kimchi* can be observed to be positioned in the region where sulfur compounds are more prevalent. Consequently, there is a slight difference in the amount or type of volatile compounds present at the early fermentation stages, showing a clear distinction on the 20th day of fermentation.

## Correlation of volatile compounds and fermentation periods

The correlation between the volatile compounds identified in *jogi-baechu-kimchi* and *baechu-kimchi* and the fermentation times was analyzed using a heatmap (Fig 3). Volatile compounds such as aldehydes, isothiocyanates, ketones, and nitriles tend to decrease or become undetectable as fermentation progresses. Various prior studies have demonstrated the reduction or absence of these components during fermentation [32, 42–44]. By contrast, acids, alcohols, isoprenoids, sulfur compounds, and terpenes are volatile compounds that are detected as fermentation progresses. Among these, the changes in volatile compounds due to *jogi* addition show significant differences, particularly in sulfur compounds. 3-Prop-2-enylsulfanylprop-1-ene, 1-(methyldisulfanyl)propane, 2-(prop-2-enyldisulfanyl)propane, 1,2,5,6,7,8-hexahydropyrrolizine-3-thione, and 1-(methyldisulfanyl)-1-methylsulfanylpropane are components that showed statistically significant results only in *jogi-baechu-kimchi*. Hence, the changes in volatile compounds due to *jogi* addition and fermentation were confirmed.

The components that consistently showed positive results in volatile compounds are acetic acid, 6-methylhept-5-en-2-ol, (*E*)-oct-2-en-1-ol, 4-isothiocyanatobut-1-ene, 1-(methyltrisulfanyl)propane, 1-(propyltrisulfanyl)propane, 3,7-dimethylocta-1,6-dien-3-ol, and (3*E*,6*E*)-3,7,11-trimethyldodeca-1,3,6,10-tetraene. These eight volatile compounds were commonly detected on the 20th day of fermentation. Among them, acetic acid, (*E*)-oct-2-en-1-ol, 1-(methyltrisulfanyl)propane, 1-(propyltrisulfanyl)propane, 3,7-dimethylocta-1,6-dien-3-ol, and (3*E*,6*E*)-3,7,11-trimethyldodeca-1,3,6,10-tetraene were also identified in previous studies examining kimchi volatile compounds [32, 35, 43, 45]. Additionally, these five volatile compounds showed a positive correlation with each other (Fig 2C). Therefore, these components could be considered to be potential biomarkers for confirming the progress of the fermentation.

To confirm the correlation among volatile compounds, Pearson correlation was analyzed as a heatmap (Fig 2C). While the results in Fig 3 were to confirm the correlation between fermentation period and volatile compounds, the results in Fig 2C were to confirm the correlation between volatile compounds. To confirm and verify the results in Fig 3, the correlation between volatile compounds was verified using PAST 4 and SPSS software (Fig 2C). 67 volatile compounds could be divided into two categories, 16 of which were negative for the rest and showed positive correlation among themselves. The volatile compounds showing negative correlation are those that are not detected in the later stages of fermentation, unlike those that are

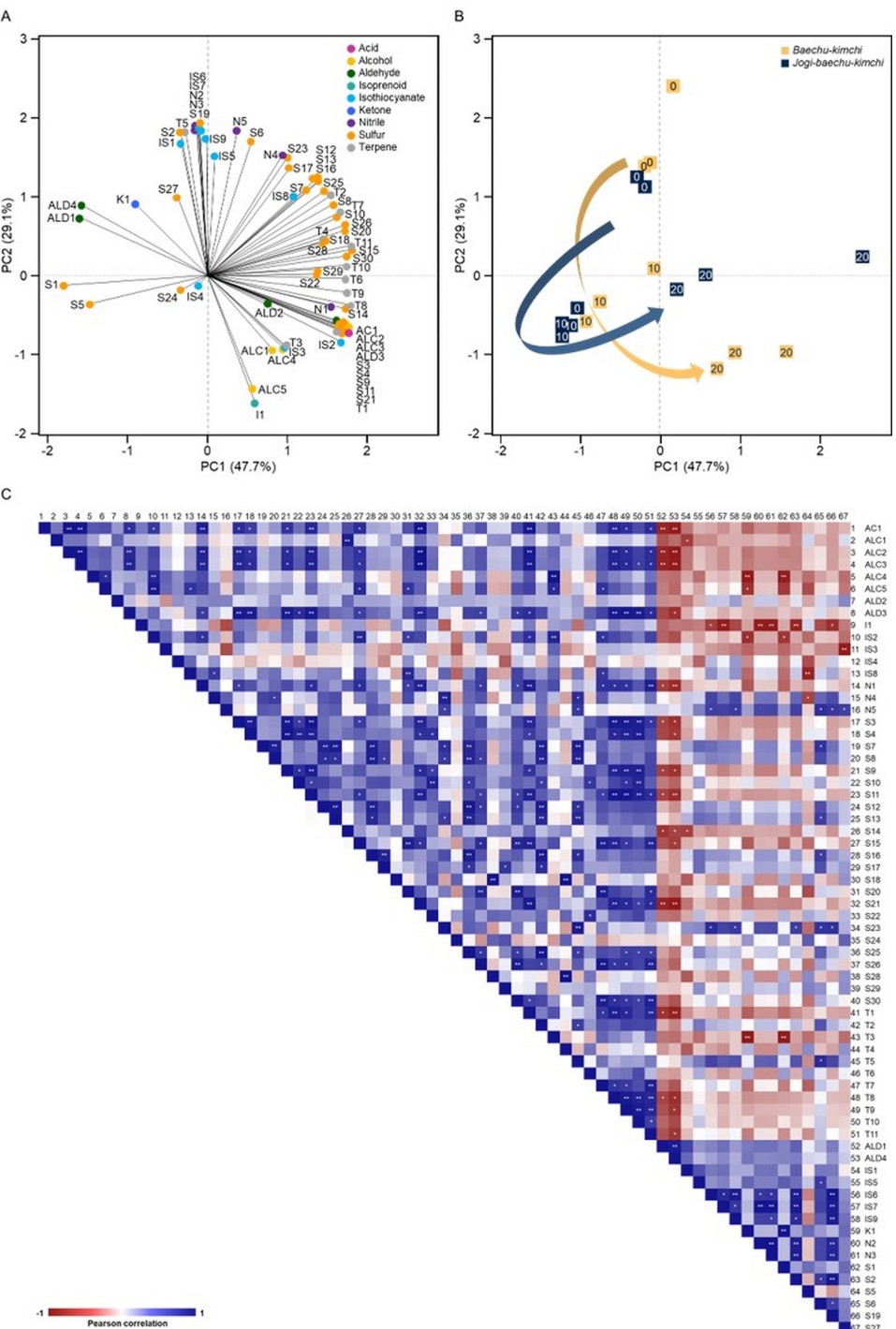

**Fig 2. Characterization of volatile compounds produced from two types of *baechu-kimch* during fermentation.**
Principal component analysis (PCA) loading plot (A), PCA factor scores (B), and Pearson correlation coefficient (C).
In (A), volatile compounds are displayed in different colors for each category. In (B), the number means the number of
days of fermentation. In (C), * and ** represent statistical significance at the 0.05 and 0.01 levels, respectively.
Abbreviations: AC, acid; ALC, alcohol; ALD, aldehyde; I, isoprenoid; IS, isothiocyanate; K, ketone; N, nitrile; S, Sulfur;
T, terpene. JBK, *jogi-baechu-kimchi*; BK, *baechu-kimchi*. The volatile compound abbreviations were added in the
Table 1.

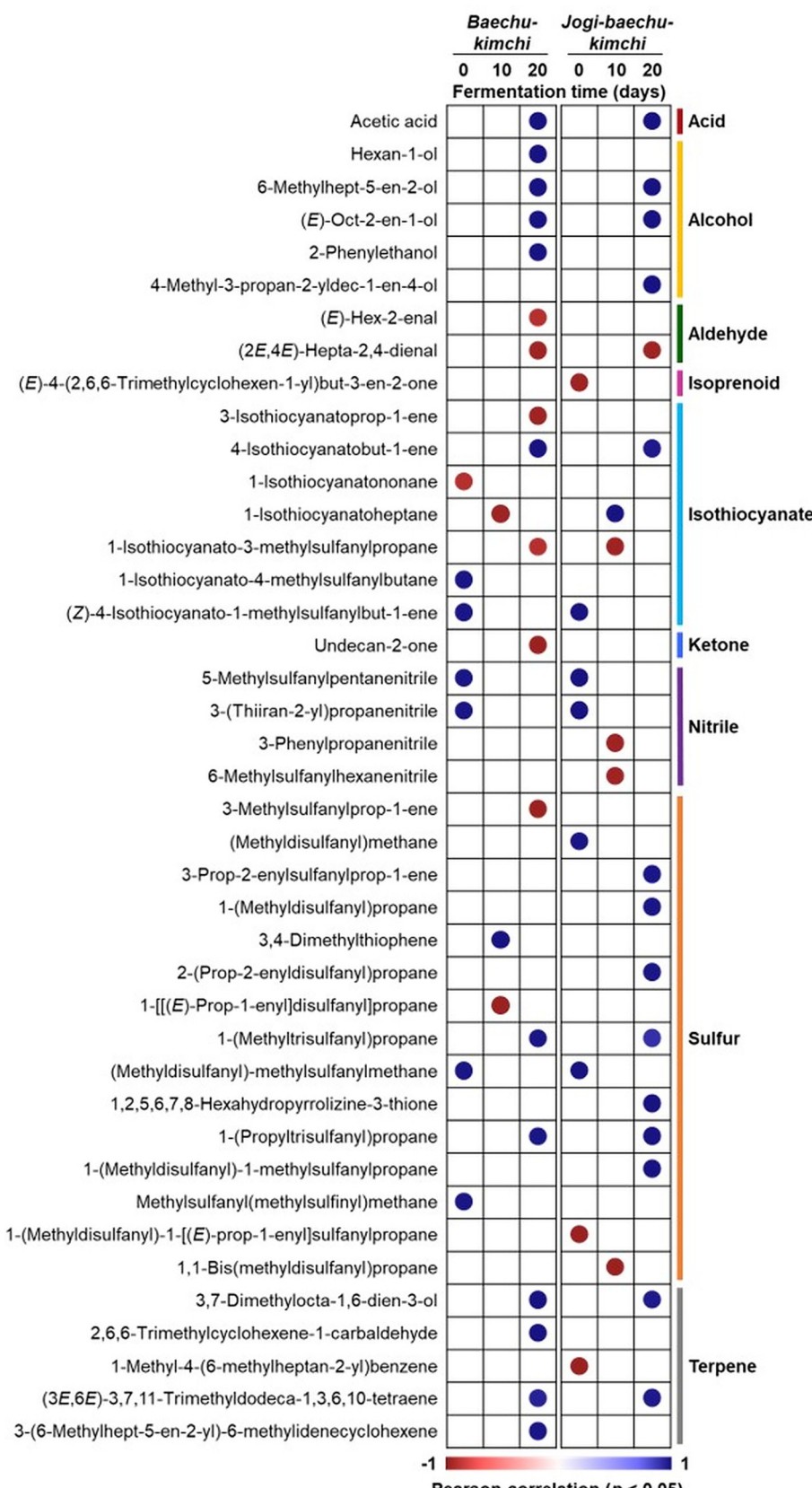

**Fig 3. Correlation between volatile compounds and fermentation periods.**

generated as fermentation progresses. Additionally, compounds with negative correlations in Fig 3 also exhibit negative correlations in Fig 2C. For example, the volatile compounds that significantly decreased in the later stages of fermentation, such as (*E*)-hex-2-enal, (2*E*,4*E*)-hepta-2,4-dienal, 3-isothiocyanatoprop-1-ene, 1-isothiocyanato-3-methylsulfanylpropane, undecan-2-one, and 3-methylsulfanylprop-1-ene, belong to the group showing negative values in Fig 2C. Therefore, the correlation between increasing and decreasing volatile compounds as fermentation progresses can be identified.

## Correlation of volatile compounds and microbial community in kimchi

In the kimchi used in this experiment, as analyzed in a previous study investigating microbial communities, *jogi* has been found to initially have a minor impact on the microbial community of cabbage kimchi but almost no influence after 10 days of fermentation [24, 25]. Nevertheless, we aimed to analyze the correlation between the microbial community and volatile compounds of kimchi (Fig 4). Because *jogi* addition had a minimal impact on the microbiota, we analyzed the correlation between the volatile compounds detected in all samples, regardless of the addition of seafood, and the microbial communities separated through a cultivation-dependent method, with the results yielding $p < 0.05$ values. Among the nine categories of volatile compounds, only five categories showed a correlation with the microbial community. *Bacillus* species exhibited a positive correlation with isothiocyanates, nitriles, and sulfur compounds, *Lactilactobacillus sakei* showed a positive correlation with isoprenoids and isothiocyanates, and *Mammaliicoccus sciuri* had a positive correlation with isothiocyanates and sulfur compounds. In contrast, *Weissella cibaria* and *Weissella koreensis*, both of which belong to the

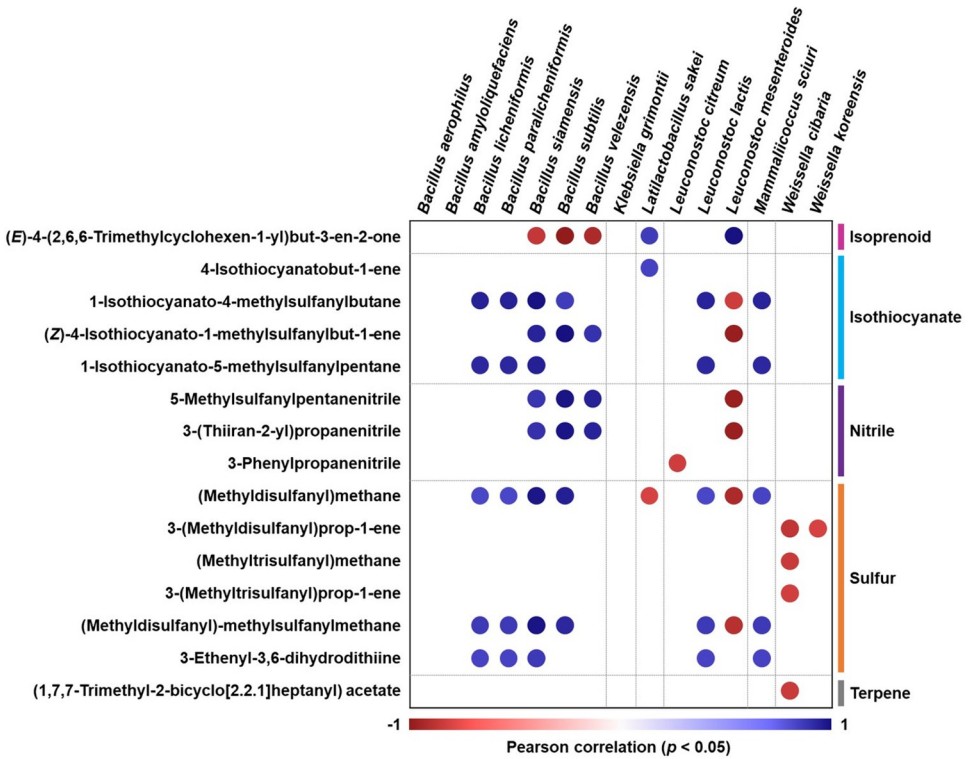

**Fig 4. Correlation heatmap between the key volatile compounds and dominant bacteria in *baechu-kimchi* and *jogi-baechu-kimchi*.**

*Weissella* genus, exhibited a negative correlation with sulfur compounds and terpenes. Meanwhile, the *Leuconostoc* genus, which was represented by three different species, showed distinct correlations, thus suggesting the existence of inter-species differences.

Interestingly, among the volatile compounds that are thought to be related to microorganisms, excluding (*E*)-4-(2,6,6-trimethylcyclohexen-1-yl)but-3-en-2-one and 4-isothiocyanatobut-1-ene, most of the volatile compounds were primarily detected in the early stages of fermentation. Moreover, the microorganisms that were positively correlated with these volatile compounds were found to proportionately decrease as fermentation progressed. These results suggest that the rich volatile compounds derived from various raw materials in the early stages of fermentation stabilize the microbial community, thus allowing the volatile compounds to harmonize as fermentation progresses. The results also indicate that the impact of lactic acid bacteria on volatile compounds is lower than that of the *Bacillus* genus. In previous studies have also shown that the microbial community changes and pH decreases during kimchi fermentation, and that volatile compounds decrease when the pH decreases rapidly [45], so it can be assumed that this is due to the effect of pH change during fermentation. However, the (*E*)-4-(2,6,6-trimethylcyclohexen-1-yl)but-3-en-2-one and 4-isothiocyanatobut-1-ene detected in the late stages of fermentation show a positive correlation with *Lb*. *sakei*, indicating its contribution to the volatile compounds in the final stages of fermentation.

## Conclusions

Our previous research found that adding *jogi* to kimchi resulted in higher levels of umami components, such as aspartic acid and glutamic acid, immediately after preparation, and that the amino acid production patterns changed by the 20th day of fermentation. However, the addition of *jogi* did not show any significant differences in organic acid content or microbial communities. Using the same kimchi from our previous experiment, we confirmed that the addition of *jogi* influenced the production of volatile compounds, particularly sulfur compounds, by the 20th day of fermentation. Five sulfur compounds were specifically attributed to the addition of *jogi*. These findings suggest that *jogi* influenced umami taste during the early stages of fermentation and contributed to the production of sulfur compounds by the 20th day. This implies that consumers may have the option to choose kimchi based on the presence of *jogi* as an ingredient or the sensory characteristics associated with different fermentation periods. Therefore, this study is expected to influence kimchi selection based on consumer preferences, allowing for more tailored choices in kimchi products.

## Author Contributions

**Conceptualization:** Gawon Lee, Sojeong Heo, Do-Won Jeong.

**Data curation:** Gawon Lee, Sojeong Heo.

**Formal analysis:** Gawon Lee, Sojeong Heo, Junghyun Park.

**Funding acquisition:** Do-Won Jeong.

**Investigation:** Gawon Lee, Junghyun Park, Jung-Sug Lee, Do-Won Jeong.

**Methodology:** Gawon Lee, Sojeong Heo, Junghyun Park, Jung-Sug Lee.

**Project administration:** Do-Won Jeong.

**Supervision:** Do-Won Jeong.

**Writing – original draft:** Gawon Lee, Sojeong Heo, Do-Won Jeong.

**Writing – review & editing:** Gawon Lee, Jung-Sug Lee, Do-Won Jeong.

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
