## [Decision Letter · Decision Letter 0]

20 Aug 2024

PONE-D-24-29960Effects of jogi, Micropogonias undulatus, addition on the production of volatile compounds in baechu-kimchiPLOS ONE

Dear Dr. Jeong,

Thank you for submitting your manuscript to PLOS ONE. After careful consideration, we feel that it has merit but does not fully meet PLOS ONE’s publication criteria as it currently stands. Therefore, we invite you to submit a revised version of the manuscript that addresses the points raised during the review process.

We look forward to receiving your revised manuscript.

Kind regards,

Gurudeeban Selvaraj

Academic Editor

PLOS ONE

Journal Requirements:

   "This work was supported by a grant (RS-2022-IP322014) of Korea Institute of Planning and Evaluation for Technology in Food, Agriculture and Forestry (IPET) through the High Value-added Food Technology Development Program funded by the Ministry of Agriculture, Food and Rural Affairs (MAFRA), Republic of Korea."

Reviewers' comments:

Reviewer's Responses to Questions

**Comments to the Author**

1. Is the manuscript technically sound, and do the data support the conclusions?

Reviewer #1: Partly

Reviewer #2: Yes

2. Has the statistical analysis been performed appropriately and rigorously? 

Reviewer #1: Yes

Reviewer #2: Yes

3. Have the authors made all data underlying the findings in their manuscript fully available?

Reviewer #1: No

Reviewer #2: Yes

4. Is the manuscript presented in an intelligible fashion and written in standard English?

Reviewer #1: Yes

Reviewer #2: Yes

5. Review Comments to the Author

Reviewer #1: Impact of addition of jogi on volatile compounds in kimchi were studies, however, it would be desirable to add details about its impact on taste and aroma? Add references, if already reported.

L95: why sampling was done every 5 days ? should not based on previously experimentation of microbila communities dynamics? Or keep in view the major shift in composition? Moreovre, results shown data at first, 10th and 20th day?

- In general results presentation is good, but discuss lack rigor and comparison.

- Table 1; the values in decimals are relative abundence or quantificaiton ? no unit mentioned.

- Figure 1 B: how the amount was measured, nothing is given in materials and methods section

- Figure; 2: title is not descriptive

- Figur 4; was correlations between individual microbes not the microbial communities and metabolic compounds, so title need correction,

The new compounds were detected because of addition of jogi but nothing is disucssion the nature of metabolic pathways involved in the production of metabolities, what if these compounds were already present in jogi and slowly released during fermentation time ?

Any metabolic analysis of raw jogi is imparative to establish wheather the production was due to addition of jogi or its provide the precursers only?

Reviewer #2: The manuscript entitled "Effects of jogi, Micropogonias undulatus, addition on the production of volatile compounds in baechu-kimchi" by Lee et al. intends to explore the effect of jogi addition on the volatile compounds presence in the kimchi. The manuscript is interesting and in the scope of Plos One, bringing new knowledge on the field of research. However, the methodology and Results and Discussion have some shortcomings. Detailed comments are provided in the highlighted text fragments.

6. PLOS authors have the option to publish the peer review history of their article (what does this mean?). If published, this will include your full peer review and any attached files.

Reviewer #1: **Yes: **Prof. Dr. Muhammad Imran

Reviewer #2: No

---

## [Decision Letter · Decision Letter 1]

8 Oct 2024

Effects of jogi, Micropogonias undulatus, addition on the production of volatile compounds in baechu-kimchi

PONE-D-24-29960R1

Dear Dr. Jeong,

We’re pleased to inform you that your manuscript has been judged scientifically suitable for publication and will be formally accepted for publication once it meets all outstanding technical requirements.

Kind regards,

Gurudeeban Selvaraj, PhD

Academic Editor

PLOS ONE

Reviewers' comments:

Reviewer's Responses to Questions

**Comments to the Author**

1. If the authors have adequately addressed your comments raised in a previous round of review and you feel that this manuscript is now acceptable for publication, you may indicate that here to bypass the “Comments to the Author” section, enter your conflict of interest statement in the “Confidential to Editor” section, and submit your "Accept" recommendation.

Reviewer #1: All comments have been addressed

Reviewer #2: All comments have been addressed

2. Is the manuscript technically sound, and do the data support the conclusions?

Reviewer #1: Yes

Reviewer #2: Yes

3. Has the statistical analysis been performed appropriately and rigorously? 

Reviewer #1: Yes

Reviewer #2: Yes

4. Have the authors made all data underlying the findings in their manuscript fully available?

Reviewer #1: Yes

Reviewer #2: Yes

5. Is the manuscript presented in an intelligible fashion and written in standard English?

Reviewer #1: Yes

Reviewer #2: Yes

6. Review Comments to the Author

Reviewer #1: author has corrected the manuscript as suggested, therefore I feel hesitation in recommending this manuscript for publication in current form.

Reviewer #2: (No Response)

7. PLOS authors have the option to publish the peer review history of their article (what does this mean?). If published, this will include your full peer review and any attached files.

Reviewer #1: **Yes: **Dr.Muhammad Imran, Professor Department of Microbiology, Quaid-i-Azam University Islamabad Pakistan.

Reviewer #2: No

---

## [Editor Report · Acceptance letter]

6 Nov 2024

PONE-D-24-29960R1 

PLOS ONE

Dear Dr. Jeong, 

I'm pleased to inform you that your manuscript has been deemed suitable for publication in PLOS ONE. Congratulations! Your manuscript is now being handed over to our production team.

Kind regards, 

on behalf of

Dr. Gurudeeban Selvaraj, PhD 

Academic Editor

PLOS ONE